# Origins of Six Species of Butterflies Migrating through Northeastern Mexico: New Insights from Stable Isotope ($\delta^2$H) Analyses and a Call for Documenting Butterfly Migrations

Keith A. Hobson [1,2,*], Jackson W. Kusack [2] and Blanca X. Mora-Alvarez [2]

1 Environment and Climate Change Canada, 11 Innovation Blvd., Saskatoon, SK S7N 0H3, Canada
2 Department of Biology, University of Western Ontario, Ontario, ON N6A 5B7, Canada;
jkusack@uwo.ca (J.W.K.); bmoraalv@uwo.ca (B.X.M.-A.)
* Correspondence: khobson6@uwo.ca

**Abstract:** Determining migratory connectivity within and among diverse taxa is crucial to their conservation. Insect migrations involve millions of individuals and are often spectacular. However, in general, virtually nothing is known about their structure. With anthropogenically induced global change, we risk losing most of these migrations before they are even described. We used stable hydrogen isotope ($\delta^2$H) measurements of wings of seven species of butterflies (*Libytheana carinenta*, *Danaus gilippus*, *Phoebis sennae*, *Asterocampa leilia*, *Euptoieta claudia*, *Euptoieta hegesia*, and *Zerene cesonia*) salvaged as roadkill when migrating in fall through a narrow bottleneck in northeast Mexico. These data were used to depict the probabilistic origins in North America of six species, excluding the largely local *E. hegesia*. We determined evidence for long-distance migration in four species (*L. carinenta*, *E. claudia*, *D. glippus*, *Z. cesonia*) and present evidence for panmixia (*Z. cesonia*), chain (*Libytheana carinenta*), and leapfrog (*Danaus gilippus*) migrations in three species. Our investigation underlines the utility of the stable isotope approach to quickly establish migratory origins and connectivity in butterflies and other insect taxa, especially if they can be sampled at migratory bottlenecks. We make the case for a concerted effort to atlas butterfly migrations using the stable isotope approach.

**Keywords:** migratory connectivity; stable isotopes; deuterium; butterfly migration

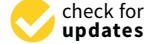



## 1. Introduction

Many animals have evolved to migrate in response to seasonal changes in resource abundance [1]. Among insects, migration is diverse and includes movements resembling dispersal, often at local scales, to intercontinental movements involving multiple generations [2]. Well-known examples of insect migration are often associated with spectacular daytime movements of Lepidoptera, Odonata, and Orthoptera, and include charismatic species (e.g., butterflies, dragonflies). Radars have also documented impressive nocturnal movements of pest species (e.g., *Autographa gamma* [3]). Indeed, in the Nearctic, one of the most impressive examples of insect migration is provided by the eastern North American monarch butterfly (*Danaus plexippus*) that annually migrates up to 5000 km from breeding grounds in the USA and Canada to discrete high-altitude overwinter colonies in oyamel fir (*Abies religiosa*) forests of central Mexico. Mated females that have successfully overwintered at these sites then return northward in the spring and initiate a series of subsequent generations that move northward in search of host milkweed (*Asclepia spp.*) plants. A final fall migrant cohort, one that has never seen the overwinter sites, then returns to those sites to complete the cycle. In the Palearctic, a similar multigenerational migration occurs between Europe and sub-Saharan Africa for the painted lady butterfly (*Vanessa cardui*), presumably timed to make use of seasonal rains in the Sahel and tropical Savannah [4]. That migratory circuit involves up to 6 generations and 4000 km. Similar intercontinental migrations are suspected for Odonata, especially the globe skimmer dragonfly (*Pantala*

*flavescens* [5–7]) and the green darner dragonfly (*Anax junius* [8]). Of course, many other Lepidoptera are known to migrate seasonally, as indicated by regular mass diurnal movements of individuals, but their origins and destinations remain largely unknown due to the difficulties in tracking small insects [2].

Recently, Satterfield et al. [9] reflected on our ignorance of insect migration in general in spite of the fact that such movements involve trillions of individuals. They reviewed the ways in which seasonal insect migrations can have important consequences for food webs, nutrient transport, pollination, and infectious disease. Importantly, they noted that, with current and predicted climate change, there is a very real possibility that seasonal insect migrations will be lost before they are documented. Clearly, there is an urgency, then, to document or describe insect migrations in concert with increased attention to declining insect numbers [10,11]. In addition, the concept of migratory connectivity been only applied to insect migration very recently [12]. This concept has important ramifications for insect conservation, especially for butterflies that typically migrate in the flight boundary layer and show a high degree of navigation ability and directed movement [13,14].

The study of animal migration has experienced quantum-level growth in recent decades, primarily due to the development of extrinsic markers that allow the tracking of individual animals at global scales [15–18]. This technological revolution has fueled an amazing array of studies showing the movements of previously poorly understood species and has advanced the field of animal movement tremendously. However, while such technological advances in tracking have been applied to smaller insects (e.g., [8,19]), in general, the study of insect migration has not benefitted from breakthroughs in individual tracking due to the obvious small size of most migratory insects. The use of intrinsic markers to infer origins, then, have been relatively better developed for insects, and the use of naturally occurring spatial patterns in genetics and stable isotope ratios ($\delta^2$H, $\delta^{18}$O, $\delta^{13}$C, $\delta^{15}$N) provides a means of inferring origins of individuals and populations [20]. This is because stable isotopes can show geographical patterns in primary production, and these isotopic patterns are passed onto higher-level consumers in predictable ways. For metabolically inert tissues such as wing chitin, stable isotope ratios are locked in once formed and the insect can be sampled at later periods or locations to infer origins through probabilistic assignment to isoscapes [21,22]. Indeed, the most well-studied migration of any insect using stable isotope analyses of wing chitin has been that of the eastern North American population of the monarch butterfly [23]. That analytical approach has since elucidated a number of ecological components of monarch migration and conservation [24–28].

Recently, Mora-Alvarez et al. [29] documented a fall migration bottleneck for monarch butterflies in northeast Mexico associated with 2 major highway crossings and predicted the annual roadkill mortality of monarchs at that site to be at least 197,000 individuals. This high mortality results from the geographic concentration of the southward bound monarchs by the Sierra Madre Oriental mountains, which forces monarchs into narrow canyons intercepted by highways. Although that study was focused on monarchs, it revealed the occurrence of massive movements of at least 36 species of other butterflies moving southward through the site. Indeed, fall migration of the family Pieridae, and especially of *Libytheana carinenta*, is well known in the states of Coahuila, Nuevo Leon, and Tamaulipas in northeastern Mexico [30–32], and local, largely anecdotal accounts have described the annual "colorful yellow and brown clouds" of butterflies (Nymphalidae and Pieridae) and other Papilionoidea families [33] through this region. However, in general, the origins and destinations of migrant butterflies at such concentration points remain entirely unknown.

We collected samples of seven species of fresh, road-killed butterfly through this bottleneck in October 2019 (*Libytheana carinenta*, *Danaus gilippus*, *Phoebis sennae*, *Asterocampa leilia*, *Euptoieta claudia*, *Euptoieta hegesia*, and *Zerene cesonia*). With the exception of the Mexican species *E. hegesia* that was intercepted in lower numbers and which was assumed to be more local, species were chosen based on their abundance, clear migration en mass

through our site, and/or potential for long-distance movements. We chose species with generally assumed broad and narrow natal latitudinal distributions in North America and used stable hydrogen isotope analyses of wing chitin to estimate natal origins. Due to the clear north-south movement of butterflies through this site, we assumed that individuals were actively migrating and could potentially represent individuals originating anywhere to the north of the site and within their known distribution ranges. Our objective was to demonstrate how the stable isotope approach can be used to quickly and effectively provide information on key aspects of butterfly movements and act as a catalyst to document migratory origins of a broad suite of species.

## 2. Materials and Methods

### 2.1. Sampling Sites

Following the authors of [29], we focused on a highway along the Sierra Madre Oriental in the state of Nuevo Leon (Highway MEX 40-D, landmark km 58, Saltillo-Monterrey; 25°39′18.54″ N, 100°27′12.24″ W). The highway dissects the natural habitat of the region and is adjacent to the high-density urban center of Monterrey. A transect (5 km) was established along sections of the highway, before and after bridges crossing canyons, as these were the most likely sites for butterfly passage. The transect included the shoulder pavement and the ditch, a width which varied depending on the highway section sampled. All dead butterflies were collected each day and placed in glassine envelopes. All samples not destined for stable isotope analyses were deposited at the Entomology Laboratory of the Facultad de Ciencias Biologicas, Universidad Autónoma de Nuevo León, México. The transect was surveyed twice per week for a total of 13 sampling days between 26 September and 20 November 2019. Specimens selected were sorted to species using identification keys [34–36] and by consulting local experts. We focused on 7 species after sorting our total sample consisting of at least 36 species. Massive migrations were witnessed throughout the study at all sites described by the authors of [29].

### 2.2. The Species

Despite the recognition that the butterfly species we analyzed are indeed migratory, almost nothing appears to be known about their movements and ultimate migratory life history. One exception is *L. carinenta*. Three subspecies were possible in our sample, but we made no attempt to distinguish among them. *L. carinenta bachmanii* occurs as far north as Canada [37] but the breeding grounds of this species are primarily south Texas and north Mexico. Massive movements of *L. carinenta* are the result of high precipitation, which increases the rapid growing of sugarberry (*Celtis laevigata*) and hackberry (*Celtis pallida*) bushes that are their host plants and are available to support large numbers of individuals. Massive migration peaks in *L. carinenta* were recorded in July, September, and October in 1971 [32]. The authors of [32] concluded that the 3 generations moved between central and south Texas and north Mexico. High numbers of *L. carinenta* were recorded in central Kansas [32] flying north in September and October but coming from Texas. However, migration occurs because the population must move and find areas that can support these large numbers of individuals to accomplish their cycle [38]. The massive migration is a unique phenomenon, which is spectacular and has only been reported in predictive years [32]. However, the population has been well represented in large numbers in the other years. In Mexico, this phenomenon has not been studied and has only been observed and recorded in popular media [30,39]. We could define the massive migration of *L. carinenta* as a short-distance migration between Mexico and Texas. However, there are *L. carinenta* breeding records in the summer in south Canada and the United States [38]. Massive migration of the subspecies *L. carinenta bachmanii* has been recorded, and droughts and rain breaks have triggered the high numbers of the population in some years in Texas [32,40].

Distributions of the remaining species have largely been described according to the authors of [35]. The species *E. claudia* has a broad breeding range centered primarily in the

southeastern United States that extends well into the east-central portion of the continent. The congeneric Mexican fritillary *E. hegesia* is centered more in distribution in Mexico. The population of *P. sennae* is more centered in the southern United States and throughout northern Mexico and is well recognized for its seasonal migrations [37,41,42]. Migration to and overwintering in Florida has been recorded [43]. The species *Danaus gilippus* occurs primarily in the southwestern United States, but the species also occurs in Florida and along the coast of the Gulf of Mexico, as well as throughout northern Mexico. Simultaneous migration of *D. gilippus* and D. plexippus in central and south Texas and northern Mexico has been recorded [29]. January records of the species in Veracruz, Mexico hint at a possible winter destination [44]. *Aterocampa leilia* has a narrow distribution in the southwestern United States, and the range is split down the northeast and northwest regions of Mexico as separated by the Sierra Madre Oriental and Sierra Madre Occidental mountain ranges. The species is resident in Arizona and central Texas [32,45]. January records in Tamaulipas, Mexico point again to potential coastal destinations [44]. *Zerene cesonia* breeds primarily in the southern United States and throughout northern Mexico but also can potentially derive from the central United States. The species cannot overwinter in Canada but is regularly recorded there [37]. It is unknown if individuals arriving north in spring to the United States are the first or second generation from Mexico. There is no evidence for spring migration but the number of individuals possibly exhibiting spring migration are so few to be below the threshold of detection.

*2.3. Stable Isotope Analyses*

Wings were soaked in a 2:1 chloroform:methanol solution, rinsed, and then dried overnight in a fume hood. Samples of $0.35 \pm 0.02$ mg of wing membrane were weighed into pressed silver $3.5 \times 5$ mm capsules and analyzed using a Eurovector Uniprep autosampler (Milan, Italy) carousel attached to a Eurovector 3000 Elemental Analyzer, coupled to a Thermo Delta V Plus isotope ratio mass spectrometer (Bremen, Germany) in continuous flow mode with helium carrier gas. After the samples were loaded, the Uniprep autosampler (heated to 60 °C) was vacuum evacuated and subsequently flushed with dry helium twice to remove adsorbed atmospheric moisture from the crushed silver capsules. The autosampler was then held under positive helium pressure for the duration of the analytical run. Two USGS keratin standards, EC-01 (formerly CBS: Caribou Hoof Standard) and EC-02 (KHS: Kudu Horn Standard of Environment Canada), were included every 10 samples. An internal laboratory standard, powdered keratin (MP Biomedicals Inc., Cat No. 90211, Lot No.9966H), was included to monitor instrument drift and provide a check on accuracy over the course of each analytical session. Samples were combusted at 1350 °C using glassy carbon. Values of $\delta^2H$ of nonexchangeable hydrogen were derived using the comparative equilibration approach of [46] and calibrated to the Vienna Standard Mean Ocean Water (VSMOW) international standard using EC-01 ($\pm$1.9‰ 1 SD, *n* = 18, accepted $\delta^2H$ = $-197.0$‰) and EC-02 ($\pm$1.6‰, *n* = 17, accepted $\delta^2H$ = $-54.1$‰). We confirmed that this approach resulted in identical assignments if new calibration standards (EC-01 accepted $-157.0$‰; EC-02 accepted $-35.3$‰) were used with correspondingly modified assignment algorithms (see below; [47]), and measurements for both sets of standards are reported in Table 1 and Supplementary Material Table S1. The overall measurement error for EC-01 and EC-02 $\delta^2H$ was ~2‰.

**Table 1.** Summary of wing stable hydrogen isotope ($\delta^2$H) (mean $\pm$SD, Vienna Standard Mean Ocean Water (VSMOW) ‰) for 7 species of road-killed butterflies moving through a migratory bottleneck highway crossing near Monterrey, Mexico in 2019. Superscript letters are the results of a Tukey post-hoc analysis whereby similar letters indicate no significant difference ($p < 0.05$). Wing $\delta^2$H values calibrated to old values (EC-01 = $-197$‰, EC-02 = $-54.1$‰) and new values (EC-01 = $-157$‰, EC-02 = $-35.3$‰) are listed.

| Species | N | Collection | $\delta^2$H (‰) Old Std | $\delta^2$H (‰) New Std |
|---|---|---|---|---|
| *L. carinenta* | 163 | 26 September to 4 November | $-94.9\pm21.6$[a] | $-68.7\pm18.2$ |
| *E. claudia* | 25 | 14 October to 20 November | $-85.7\pm13.4$[b] | $-62.8\pm11.4$ |
| *D. gilippus* | 100 | 14 October to 20 November | $-82.9\pm10.7$[b] | $-59.6\pm9.1$ |
| *Z. cesonia* | 73 | 26 September to 20 November | $-81.1\pm11.4$[b] | $-58.5\pm9.7$ |
| *A. leilia* | 38 | 8 October to 2 November | $-70.42.5\pm6.3$[c] | $-50.9\pm5.4$ |
| *E. hegesia* | 13 | 8 October to 4 November | $-69.8\pm9.4$[c] | $-49.1\pm8.1$ |
| *P. sennae* | 65 | 4 October to 20 November | $-65.16.0\pm13.2$[c] | $-46.6\pm11.9$ |

*2.4. Assignment to Origin*

We depicted the origins of wild monarchs caught in Mexico using a likelihood-based assignment method [48–50], using the wing chitin $\delta^2$H isoscape ($\delta^2$H$_w$) and an amount-weighted precipitation-to-wing calibration algorithm ($\delta^2$H$_w = 0.78 * \delta2$H$_p - 77.4$) described by the authors of [28]. This was used to convert amount-weighted mean annual precipitation $\delta^2$H ($\delta^2$H$_p$) isoscapes [51,52] into $\delta^2$H$_w$ isoscapes. We used the 9.3‰ residual SD error from this regression in our assignments [29]. We then created a spatial layer representing the known geographic range of the breeding populations of each butterfly species from Brock and Kaufman [35] and used this as a mask (i.e., clip) to limit our assignment to origin analyses.

We estimated the likelihood that a cell (pixel) within the $\delta^2$H$_w$ isoscape represented a potential origin for a sample using a normal probability density function to estimate the likelihood function based on the observed $\delta^2$H$_w$. Thus, we depicted the likely origins of each butterfly by assigning individuals to the $\delta^2$H$_w$ isoscape one at a time. We arbitrarily chose a 2:1 odds ratio to include only those pixels (coded 1) with at least a 67% probability of origin vs. all others (coded 0). This resulted in a map of binary values for each assigned individual representing presence (1) or absence (0) for each cell in the isoscape. We then summed the results of individual assignments by stacking the surfaces. We made geographic assignments to origin using functions within the R statistical computing environment v. 4.0.3 [53] within RStudio v. 1.3.95 [54] using the 'raster' v. 3.3-13 [55], rasterVis v. 0.48 [56], sp v. 1.4-4 [57], and sf v. 0.9-6 [58] packages. Thus, the final assignment surface depicted the number of individuals co-assigned at each pixel based on the odds criteria.

**3. Results**

Results of $\delta^2$H analyses of wing chitin from the seven species of butterflies we sampled are presented in Table 1. As expected, we found isotopic differences among species (ANOVA F$_{6,470} = 35.9$, $p < 0.0001$), but also clearly identified two groups of species corresponding approximately to long-distance migrants (*L. carinenta*, *E. claudia*, *D. glippus*, *Z. cesonia*) and short-distance migrants (*A. leilia*, *E. hegesia*, *P. sennae*; Table 1, unequal variance *t*-test: *t* = 14.8, *p* < 0.0001).

With the exception of the Mexican species *E. hegesia*, whose range is restricted primarily to Mexico, we assigned all species (i.e., *n* = 6 species) to probability of origin surfaces within their range. Summary isotopic data are provided in Table 1 that indicate a range from more northern origins (i.e., more negative $\delta^2$H$_w$ values) to closer origins (indicated by more positive $\delta^2$H$_w$ values). These data are followed by individual species accounts. In all cases, we note that potential origins usually include northwestern Mexico. We suggest this is simply due to the ambiguous nature of the underlying precipitation isoscape in this region, which is similar to that of northeastern Mexico. For completeness, we included the whole of northern Mexico in our depictions. However, biologically, we consider the northwestern

region of Mexico to be a highly unlikely origin for all species due to the occurrence of the Sierra Madre mountains.

### 3.1. Libytheana Carinenta

*Libytheana carinenta* had the most negative average $\delta^2H_w$ values, indicating that this species has most northern origins among the species we studied. Indeed, some individuals clearly were consistent with origins from the northernmost extent of the range (Figure 1).

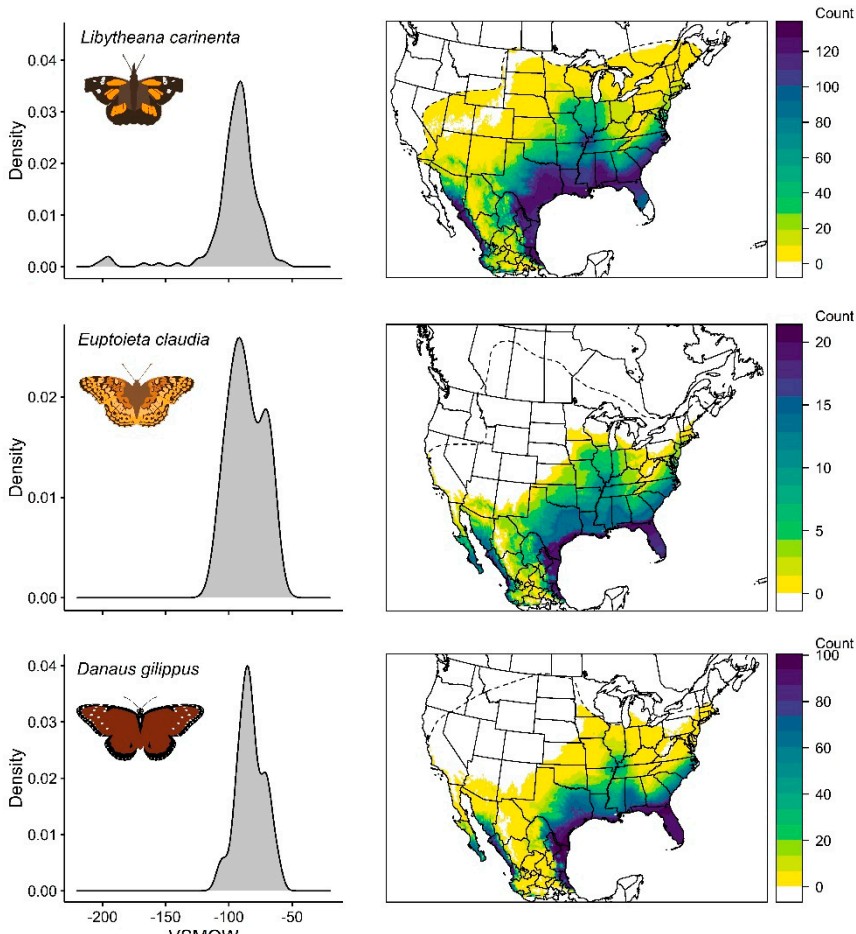

**Figure 1.** Probabilistic assignment to origin for three (longer-distance) species of butterfly intercepted during fall migration in northeast Mexico, 2019. The figure key refers to the number of individuals assigned per pixel according to a 2:1 odds criterion (see Methods).

However, the majority of the assignment was consistent, with origins generally south of the Great Lakes and throughout the southeastern USA as generally reported in the literature. The sample size of this species and the long duration of sampling also allowed us to examine the pattern of migration through the site. After removing four outliers with $\delta^2H_w$ values between $-195‰$ and $-200‰$ (i.e., indicating origins well north of the expected breeding range), we found a negative correlation between $\delta^2H_w$ values and the day of sampling (Figure 2, $p < 0.0001$), which suggests a chain migration pattern with more northern-produced butterflies arriving later than more southern-produced individuals.

### 3.2. Euptoieta Claudia

*Euptoieta claudia* also showed potential origins well north of northern Mexico and primarily in the southeastern United States. The species also showed a possible finger-like extension west of the Appalachians through to the southern Great Lakes, corresponding largely with the Mississippi River drainage (Figure 1).

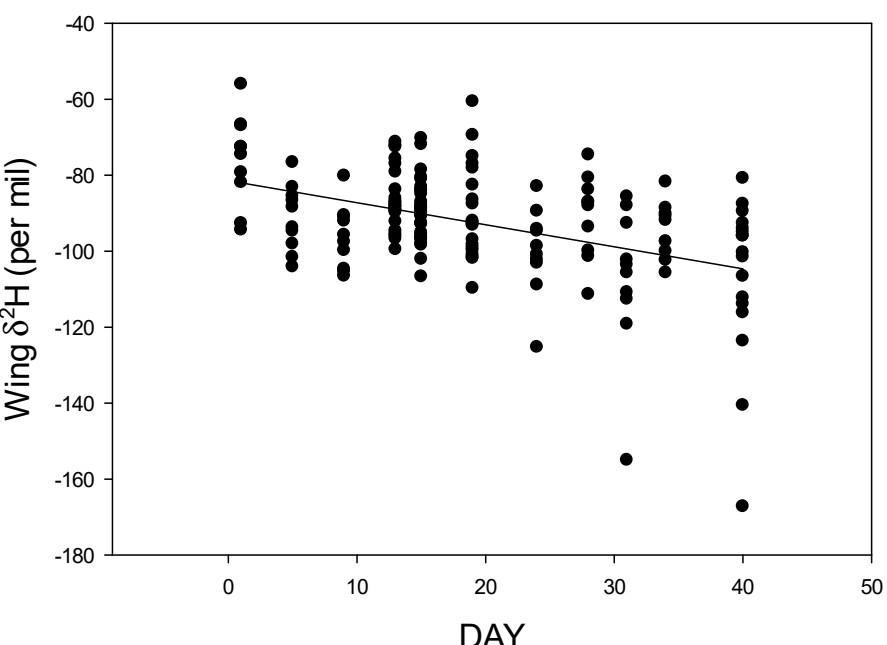

**Figure 2.** Relationship between wing $\delta^2$H and day of collection for *Libytheana carinenta* intercepted in fall migration 2019 in northeast Mexico. Y = −0.58X − 81.4, r$^2$ = 0.2. Day 0 is 26 September.

### 3.3. Danaus Gilippus

*Danaus gilippus* showed similar potential origins to the other two highly migratory species, *L. carinenta* and *E. claudia*, with a concentration to the southeastern portion of the range and an extension through the Mississippi River Valley to the southern Great Lakes (Figure 1). The large sample size and the long period of collection allowed us to examine the effect of sampling date on $\delta^2$H$_w$ values. We found a weak but significant increase in $\delta^2$H$_w$ values with time (Figure 3), suggesting either a pattern of migration with a latitude of origin or possibly more northern butterflies tending to arrive first.

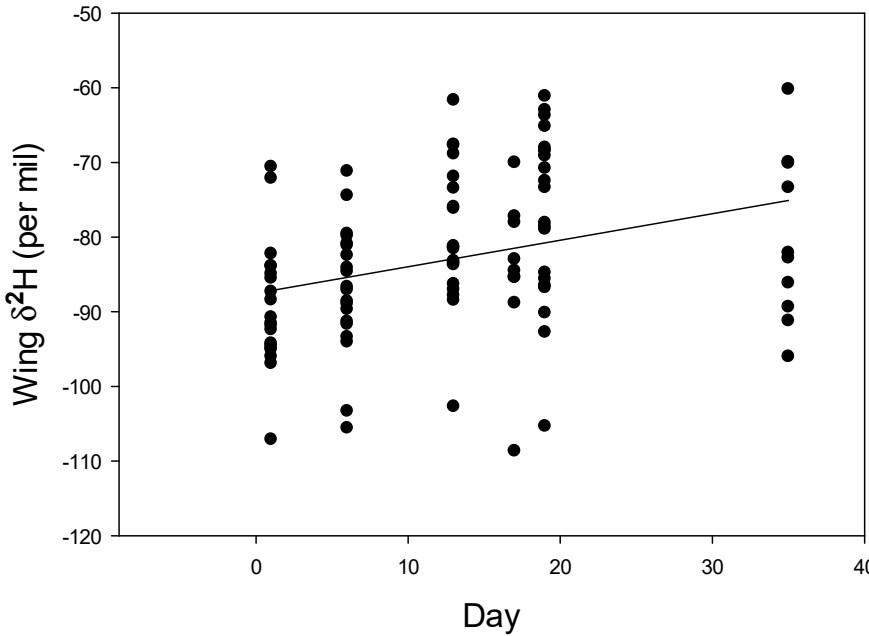

**Figure 3.** Relationship between wing $\delta^2$H and day of collection for *Danaus gilippus*. Intercepted in fall migration 2019 in northeast Mexico. Y = 0.35X − 87.5. r$^2$ = 0.11. Day 0 is 14 October.

### 3.4. Zerene Cesonia

This species showed a remarkable similarity in potential origins to those described above (Figure 4, Table 1). We found no relationship between $\delta^2H_w$ values and day of collection (slope = 0.03, $r^2$ = 0.007, $n$ = 73).

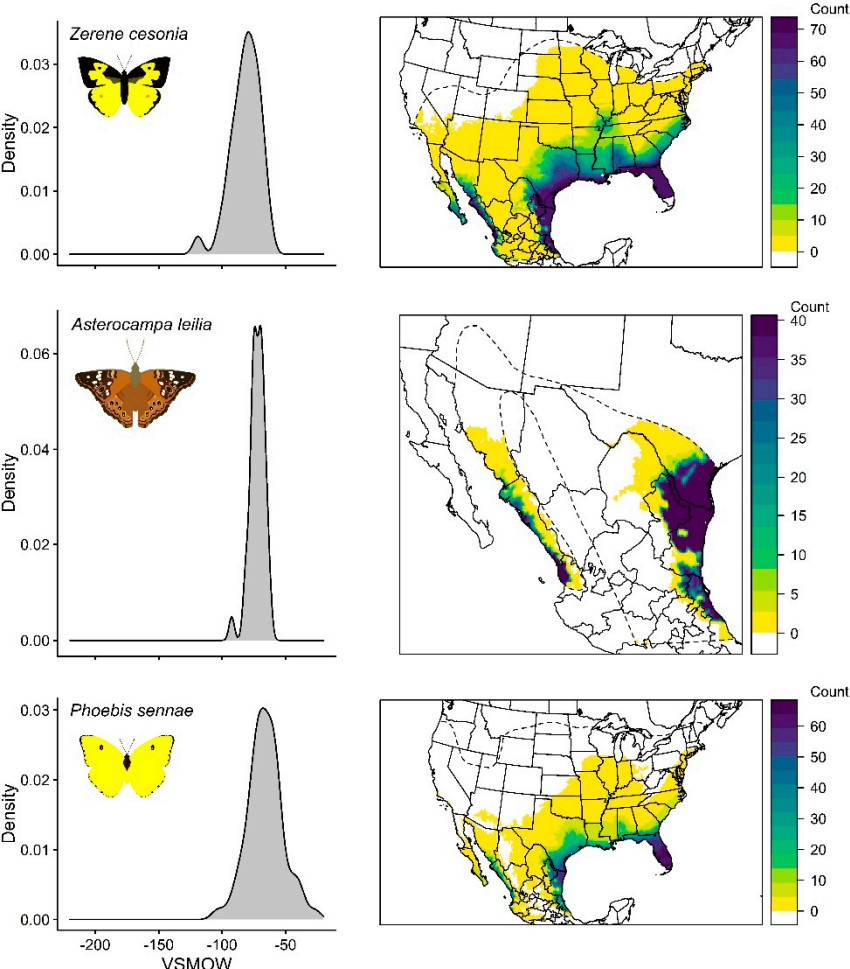

**Figure 4.** Probabilistic assignment to origin for three (shorter-distance) species of butterfly intercepted during fall migration in northeast Mexico, 2019. The figure key refers to the number of individuals assigned per pixel according to a 2:1 odds criterion (see Methods).

### 3.5. Asterocampa Leilia

Despite large movements of this species through our study site, our findings clearly indicate that this species has little chance of originating north of Mexico, as confirmed by the more positive $\delta^2H_w$ values (Figure 4, Table 1). As such, isotopic results for this species confirm a status of short-distance migration, primarily within Mexico and adjacent Arizona [45].

### 3.6. Euptoieta Hegesia

Due to the low sample size of this species ($N$ = 14) and its narrow window of sampling, we did not produce an assignment to the origin map. Nonetheless, the species served as a useful "control," with expected higher $\delta^2H_w$ values (mean $-70.2 \pm 9.3$‰) consistent with more southern (Mexican) origins (Table 1) *Phoebis sennae.*

This species was the second-most enriched in $^2H$ (Table 1), largely suggesting that individuals were derived from southern latitudes, including northeast Mexico. However, the population sampled at our site showed potential origins also around the Gulf states to the northeast of our site.

## 4. Discussion

Following previous studies on monarch butterflies in North America, our investigation demonstrates the utility of sampling butterflies at a fall migratory bottleneck in northeast Mexico as a means of inferring their latitudinal origins through the measurement of wing $\delta^2$H values. This approach is well suited to quickly determine the structure of migratory butterfly populations by making use of the substantial mortality that occurs at highway crossings, as demonstrated by Mora Alvarez et al. [29]. Without a priori expectations, we determined that four of the seven species examined (*L. carinenta*, *Z. cesonia*, *E. claudia*, *D. gilippus*) included individuals that can be considered long-distance migrants, with individuals travelling throughout their range in the United States and southern Canada and hence reflecting distances of up to several thousands of kilometers. We suggest that these species would be useful candidates for more in-depth measurements of wing size as this has been associated with distance of migration (e.g., [25,27]). The remaining three species (*P. sennae*, *A. leilia*, *E. hegesia*) were clearly migrating south en mass through our study site even though their origins were much more local. For three species, we were additionally able to evaluate patterns of movement through our site relative to latitude of origin. Here, our results were consistent with migratory panmixia (*Z. cesonia*), chain (*Libytheana carinenta*), and leapfrog (*Danaus gilippus*) migrations as has been described for other taxa [1]. Of course, other explanations are possible, and we note that the regressions between wing $\delta^2$H and date of passage were relatively weak.

The impressive fall movement of butterflies through northeast Mexico is virtually unstudied. Thus, our research is but a first step in describing migration across a subsample of species. Indeed, while we identified the approximate origins of seven species, we, of course, have no idea of their ultimate destinations. Anecdotal evidence points to possible destination in the northern Mexican coastal states, especially Tamaulipas [59]. In addition, we cannot currently distinguish between regular migration per se and dispersal related more to climate and resource conditions [2]. We strongly recommend field censuses that are designed to identify potential overwinter destinations of these and other species so that true migratory connectivity can be established through the complimentary isotopic measurement of wintering individuals (e.g., [4,27,60,61]).

We clearly demonstrated that the stable isotope approach to delineating butterfly origins in North America provides a particularly convenient tool to quickly document origins and migration in species with poorly known life history. This is because isoclines of precipitation $\delta^2$H, and hence butterfly wing $\delta^2$H, follow a particularly well-established latitudinal gradient (see figure in [61]), a phenomenon that has led to the isotopic tracing of origins for numerous taxa [21]. Insects that concentrate at migratory bottlenecks such as the species used by us in northern Mexico are especially amenable for assignment to an isotopic atlas. The salvaging of fresh road-killed individuals at such sites also removes the need to sacrifice live individuals and potentially provides a large sample for assay. In addition to continuing sampling of the numerous species of butterflies passing through migratory bottlenecks in autumn, sampling of migrant butterflies during the winter months in Mexico and possibly Central America and the Caribbean would be especially useful to link breeding and wintering sites [62]. We see this call for action as an essential component of the recent concern expressed by Satterfield et al. [9] that we stand to lose valuable migratory insect populations long before they are even described.

**Supplementary Materials:** The following are available online at https://www.mdpi.com/1424-281 8/13/3/102/s1, Table S1: Supplementary data for all butterflies sampled near Monterrey Mexico in 2019.

**Author Contributions:** Conceptualization, K.A.H.; Methodology, K.A.H., B.X.M.-A., J.W.K.; Statistical Analysis, K.A.H. and J.W.K.; Writing—Original Draft Preparation, K.A.H.; Writing—Review & Editing, K.A.H., B.X.M.-A., J.W.K.; Visualization, K.A.H. and J.W.K.; Supervision, K.A.H. and B.X.M.-A.; Project Administration, K.A.H. and B.X.M.-A.; Funding Acquisition, K.A.H. All authors have read and agreed to the published version of the manuscript.

**Funding:** This work was supported by an NSERC Discovery grant to KAH (2017-04430).

**Informed Consent Statement:** Not applicable.

**Data Availability Statement:** The data presented in this study are available in supplementary material here.

**Acknowledgments:** Special thanks to our field assistant Carlos Carrera-Treviño for his excellent assistance and Rocio Treviño Ulloa for her dedication and contribution to recording of butterfly sightings in the Correo Real Program which was supported by many volunteers. We thank Jose Davila Paulin for his insights on the movements of butterflies and help with surveys in the north of Coahuila. We thank two anonymous reviewers for their helpful comments on a previous version of the manuscript.

**Conflicts of Interest:** The authors declare no conflict of interest.

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
