# Peer review of "Origins of Six Species of Butterflies Migrating through Northeastern Mexico: New Insights from Stable Isotope (δ2H) Analyses and a Call for Documenting Butterfly Migrations"

_diversity, doi:10.3390/d13030102_

Round 1
Reviewer 1 Report
This study takes advantage of carcasses found at a migration bottleneck to investigate the origin of butterflies suspected of migratory behavior. The results are interesting and important. However, there are some minor issues, see below, that should be addressed before accepting this for publication.
Line 190: Please include a description for acronyms like VSMOW. It’s challenging with a deeply technical method like stable isotopes to know how much background information to provide for readers unfamiliar with the jargon. Because you use this term in Table 1, and thus it is presumably important to understanding the results, please add a sentence or so describing the meaning and how to interpret it.
Figs 1 & 3: The color use in these figures is confusing. I’m guessing that the color of the area under the curve on the left is the number of individuals, but the key seems to be more related to the maps on the right, which imply a spread of values.
Line 255: I don’t understand why the pattern in Fig 2 describes chain migration. To me it just suggests that butterflies traveling further arrived later, which is still an interesting clue; departure cue may not vary with distance/latitude. Also interesting that this pattern is reversed with Danaus gilippus. This also applies to lines 319-321; the D. gilippus pattern was not leapfrog migration either. In either case, you would not see members of the whole population passing through the same point. If you want to describe these patterns of migration as chain and leapfrog, please add examples and background on this in the introduction as well.
Line 270: I don’t think you mean to cite figure 3 here.
It would be helpful to have the order of species accounts in the Results section be the same as the order in Table 1.
Reviewer 2 Report
This is an excellent and data-rich paper exploring at an unprecedented scale the migration of multiple species of butterfly from a bottleneck site. I like this paper and I think the study will encourage people to do more research using isotope to map migratory patterns in insects. So I strongly recommend this manuscript for publication. The manuscript is also very well-written, the figures are of good quality, and the interpretation is well-referenced.
I do have to recommend moderate revisions, however, because of the issues below associated with the methodology:
- I don’t fully understand the argument for not recalibrating the data to the revisited standard values (Soto et al.)? I understand that the authors want to be consistent with previous work but I think the proper approach would be to recalibrate all the past values to the recalibrated standards… I don’t think perpetuating datasets measured at different reference scales in the literature is good. My fear is that, for most readers, the details of the analysis of exchangeable H are unknown. So it will be unclear to them that these d2H data are not comparable to those generated in other labs. So I recommend recalibrating everything to Soto et al. accepted standard values.
Additionally I think the sentence: “We used these former calibration standard values because our original assignment algorithms were based on these (Hobson et al 1999)” is not fully correct. My understanding is that while Hobson et al. 1999 is a fantastic paper, the analytical approach used there cannot be recalibrated to modern standards because the standards and approach used at the time were not tested by Soto et al. So I think the proper citation in that sentence would Hobson et al. 2019. - AssignR provides a good workflow to generate tissue-specific isoscape and their uncertainty. So as you have to recalibrate the values as mentionned in my first point, why not using that function instead of using a fixed error of 9.3 per mile. It will provide a spatially explicit uncertainty.
- In the SI it would be good to add analytical uncertainty for each sample. Also add two d2H columns one with the raw measured values and one with the recalibrated values (to Soto et al.). Other metadata available could also be incorporated to make those easily transferable to a database?
- To make this study fully reproducible, I also recommend adding the R script used for generating the isoscape and the assignments.
